# Resonant X-ray emission spectroscopy from broadband stochastic pulses at an X-ray free electron laser

Franklin D. Fuller [1✉], Anton Loukianov[1], Tsukasa Takanashi[2], Daehyun You[2], Yiwen Li [2], Kiyoshi Ueda[2], Thomas Fransson[1], Makina Yabashi [3], Tetsuo Katayama [3,4], Tsu-Chien Weng[5], Roberto Alonso-Mori [1], Uwe Bergmann[6], Jan Kern [7], Vittal K. Yachandra[7], Philippe Wernet [8] & Junko Yano [7]

Hard X-ray spectroscopy is an element specific probe of electronic state, but signals are weak and require intense light to study low concentration samples. Free electron laser facilities offer the highest intensity X-rays of any available light source. The light produced at such facilities is stochastic, with spikey, broadband spectra that change drastically from shot to shot. Here, using aqueous ferrocyanide, we show that the resonant X-ray emission (RXES) spectrum can be inferred by correlating for each shot the fluorescence intensity from the sample with spectra of the fluctuating, self-amplified spontaneous emission (SASE) source. We obtain resolved narrow and chemically rich information in core-to-valence transitions of the pre-edge region at the Fe K-edge. Our approach avoids monochromatization, provides higher photon flux to the sample, and allows non-resonant signals like elastic scattering to be simultaneously recorded. The spectra obtained match well with spectra measured using a monochromator. We also show that inaccurate measurements of the stochastic light spectra reduce the measurement efficiency of our approach.

[1] SLAC National Accelerator Laboratory, Menlo Park, CA, USA. [2] Tohoku University, Sendai, Miyagi, Japan. [3] RIKEN SPring-8 Center, Sayo, Hyogo, Japan. [4] Japan Synchrotron Radiation Research Institute, Sayo, Hyogo, Japan. [5] School of Physical Science and Technology, ShanghaiTech University, Shanghai, China. [6] University of Wisconsin, Madison, WI, USA. [7] Lawrence Berkeley National Laboratory, Berkeley, CA, USA. [8] Uppsala University, Uppsala, Sweden. ✉email: fdfuller@slac.stanford.edu

Over the last 10 years, X-ray Free Electron Lasers (XFELs) have shown their potential for being transformative tools to study chemical and structural dynamics of materials, molecular and biological systems. The X-ray pulses generated by an XFEL are extremely intense with short pulse durations of typically 50 fs or less and they contain $10^{12}$ to $10^{13}$ photons per pulse, which are as many photons as synchrotron radiation sources generate in one second. Due to its high spatial coherence, XFEL radiation can also be easily focused into a micrometer-size spot. These exceptional properties of XFEL radiation, together with shot-by-shot data collection, make it possible to follow chemical reactions, transformations of materials, molecular dynamics and biological processes in real time[1–3].

Except for seeded XFELs[4], XFEL sources are based on the principle of self-amplified spontaneous emission (SASE) for the amplification of short X-ray pulses. SASE XFELs are intrinsically stochastic and produce pulses with spikey spectral amplitude that change dramatically from shot to shot[5]. When tuned to an atomic edge, the SASE spectral amplitude variation induces shot to shot fluctuations in excited state population that correspond to a weighted superposition of populations created by the constituent colors of the SASE pulse. Correlating SASE spectral amplitude with a signal proportional to excited state population, therefore allows one to recover various resonant signals like X-ray absorption, anomalous scattering, and Resonant X-ray Emission Spectroscopy (RXES), with spectral resolution determined by the SASE spikes, rather than the overall SASE bandwidth. In order to recover a spectrum, i.e., the monochromatic response, from polychromatic stochastic light, the spectrum of the source must be measured accurately for each shot along with every observation from the sample so that the desired signal can be recovered from the correlation of the two measurements. Hard X-ray (>5 keV) signals are well suited to explore using SASE spectral fluctuations for resonant measurements, since the available SASE spectral diagnostics[6–9] have both high spectral resolution and excellent signal to noise characteristics.

Kayser et al.[10] demonstrated for the first time that hard X-ray absorption spectra can be experimentally obtained by correlation of X-ray emission with measurements of the stochastic XFEL pulse spectrum. They were able to reconstruct Fe K-edge absorption of $Fe_2O_3$ nanoparticles. The recovered spectra of the $Fe_2O_3$ nanoparticles resolved the absorption edge and evidenced non-linear effects, but not finer spectral details like pre-edge features. The current study is focused on resolving such features, since stochastic detection of the pre-edge region, which is ~10× weaker than the main edge, greatly broadens its utility in revealing essential chemical information in biological, chemical and material science systems. The weak features of the K pre-edge region are a desirable probe of the chemical environment of the absorbing metal, as they are mainly comprised of transitions from core (1 s) orbitals to valence orbitals and thus report on the valence state of the metal. The K pre-edge is sensitive to the metal's oxidation state, ligand coordination and symmetry, and the type of ligand bound to it. 1s2p ($K_\alpha$ detected) Resonant Inelastic X-ray Scattering (RIXS) signal in the RXES pre-edge region contains more chemical information than X-ray absorption alone. In 1s2p RIXS, the fluorescence from the sample at various incidence energies is collected, resulting in a two-dimensional spectrum that reveals a coupling between the distribution of final states involved in emission and pre-edge resonances involved in absorption. The coupling can manifest as a shift or splitting of pre-edge peaks along the energy transfer direction, i.e., the difference between incident and emitted energy, and thus contains unique information not resolved in the emission or absorption spectra alone[11].

The prospect of using SASE light for resonant experiments is exciting because the bandwidth mismatch between the

monochromator and source is avoided, increasing the number of photons that can be delivered to the sample for each XFEL shot by up to 100 times. The commensurately increased signal yield of SASE light can result in faster acquisition times when the signal is close to the noise floor. The potential benefits of polychromatic light relative to monochromatic light for improving spectroscopic measurement times has been known for a long time in the optical and infrared wavelengths, particularly in Fourier spectroscopy[12] and more recently in compressed sensing[13]. We examine the possibility of faster acquisition times via direct experimental comparison as well as via simulation and conclude that a measurement time advantage using SASE light is possible in some experiments. The increased number of available photons would also make combined time resolved resonant spectroscopy and scattering experiments possible. In combined time resolved scattering/spectroscopy experiments, the high intensity afforded by SASE pulses is crucial to the experimental viability of the scattering signal, so only spectroscopies which can be performed with a SASE beam like X-ray emission have been paired with them to date. Such emission/scattering experiments have been used[14–17] to observe concurrent atomic motions via scattering and localized valence electronic state evolution via emission spectroscopy. Stochastic resonant spectroscopies would complement the valence electronic state information that can already be obtained through emission spectroscopy. Beyond the traditional linear spectroscopy regime, the approach we describe here to accurately correlate incoming broadband X-rays and outgoing signals will enable non-linear spectroscopies by increasing power on the sample, while also allowing shorter pulses than a monochromator permits[18]. Shorter X-ray pulses are needed to probe attosecond processes that are fundamental to electron-light interaction and electron-electron correlation.

The central focus of this work is to examine the viability of RIXS spectroscopy with stochastic XFEL pulses for the study of chemical systems and to determine where improvements can be made. We examine the spectral details which stochastic spectroscopy can experimentally resolve and directly compare the recovered signal to monochromatic measurements using the same setup and analysis approach. In this comparison we find good agreement between the two approaches that improves as measurement time increases. For short measurement times, however, we see some discrepancy between the two approaches. We examined the differences via simulation and show how signal recovery is affected by measurement noise in the SASE diagnostic. We conclude that SASE diagnostic noise can reduce measurement time efficiency for stochastic spectroscopy and explain some of the differences we see between stochastic and monochromatic measurements. Finally, via simulation, we examine the impact XFEL statistics and fluorescence detection noise (and by proxy sample concentration) has on the measurement efficiency of stochastic spectroscopy. We find that stochastic spectroscopy scales just as well as monochromatic spectroscopy with respect to sample concentration and that increasing the randomness of the XFEL spectrum improves stochastic spectroscopy performance.

## Results

**Comparison of monochromatic and stochastic measurements**. A comparison of 1s2p RIXS planes of aqueous ferrocyanide reconstructed from both monochromatic and polychromatic SASE excitation for the same measurement time is shown in Fig. 1a. The two RIXS reconstructions qualitatively agree with one another. Notably, both can resolve a well-known pre-edge feature at around 7.113 keV, with the monochromatic measurement peaking at slightly lower incidence energy than we see for the polychromatic SASE beam. Supplementary Figure 1 examines the stochastic

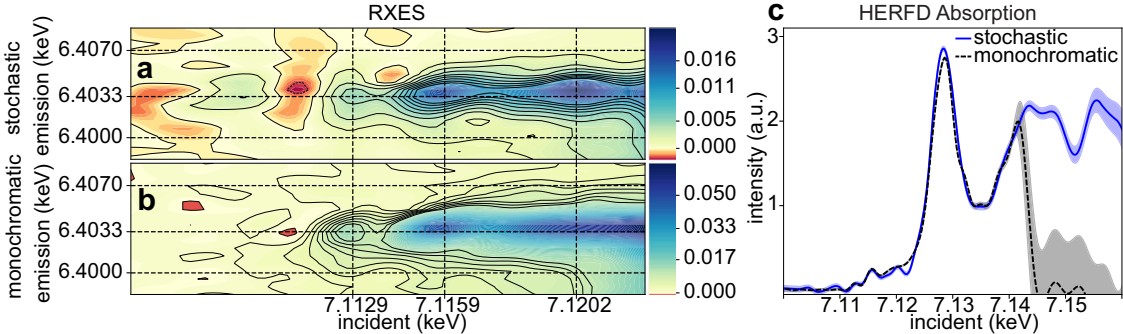

**Fig. 1 Experimental comparison of stochastic and monochromatic measurements.** The recovered Fe 1s2p RIXS planes using the polychromatic SASE beam is shown in panel **a** while the same obtained with a monochromatized SASE beam is shown in panel **b**. The spectra were recovered from 42,000 shots corresponding to ~24 min of measurement time at 30 Hz in both cases. Solid black contour lines indicate an increase of RIXS intensity by 2σ, where σ is the standard deviation of the predictive distribution estimated for a given emission/incident energy point. These contour lines can be treated as an estimate of relative significance, where peaks of interest are separated from surroundings by many contours. A slice at constant emission energy (6.4033 keV) is shown in panel **c** over a much larger incidence energy range. For both measurements, the XFEL central energy is fixed and scanning of the mono was limited to 7.100–7.140 keV. The SASE spectral intensity is centered on the Fe pre-edge region and tails off above 7.130 keV. Error bars in panel **c** likewise indicate 2σ bounds of the solution.

reconstruction with around 2.5 times more shots and we see that this gives an improved agreement between monochromatic and stochastic measurements, mainly in the vicinity of 7.120 keV. Both monochromatic and stochastic measurements exhibit a RIXS shift of the pre-edge peak in Fig. 1a, where the peak appears at slightly lower emission energy than the emission does for higher incident photon energies. In the edge region from 7.116 to 7.124 keV (Fig. 2b) we see again qualitative agreement, with noticeable differences in peak location around 7.120 keV, but good agreement in peak location is seen at 7.116 keV. Black contours in the figures are drawn at multiples of the standard deviation estimation from our fit (see Methods). Judged by the number of contour lines, the monochromatic measurement has higher contrast than the stochastic measurement with respect to the estimated variance for the pre-edge peak feature at 7.113 keV. This relative confidence in the pre-edge peak intensity for the stochastic reconstruction does improve slightly with added data, as shown in Supplementary Fig. 1a, but not to the level seen with a monochromatic beam.

A scaled slice of the RXES plane at constant emission energy 6.4033 keV from the SASE beam measurements is overlaid with that from the monochromatic beam in Fig. 1b across the near edge region (7.100–7.160 keV). This single emission energy ($K_{\alpha 1}$)-detected X-ray absorption is also referred to as High-Energy Resolved Fluorescence-detected (HERFD) absorption[19] and the sharper spectral features it gives relative to X-ray absorption make it a more incisive probe of valence state[20]. The monochromatic energy scan was terminated around 7.140 keV, as indicated by the sharp drop there, while the stochastic signal recovery extends further in energy, covering the same spectral bandwidth as the SASE beam. Where the SASE spectral intensity is weak on average, the uncertainty of the fit becomes higher, as indicated by the shaded 95% confidence interval. For the monochromatic measurement, the spectral content in the SASE diagnostic above 7.140 keV is just noise, which is why the uncertainty reported in the monochromatic spectrum is so high there. The agreement between HERFD spectra at energies above 7.120 keV is quite good until the SASE spectral intensity becomes weak. From the HERFD comparison, we see that stronger absorption signals are well handled by the stochastic spectroscopy approach so chemical analysis relying on the near-edge region[21,22] is feasible. Note that the monochromatic signal has been corrected for a spatial intensity profile effect due to slight deviation of the beam on changing from offset mirrors to the double crystal monochromator, which is described in Supplementary Fig. 2.

**Probing systematic effects in the SASE diagnostic signal.** That the Fe pre-edge feature is recoverable from the stochastic measurement and that it, along with other spectral features, is largely consistent with that of the monochromatic measurement indicates stochastic spectroscopy has suitable sensitivity to be broadly useful in the study of chemical systems. However, the differences of peak location and shape between the two measurements, particularly in the vicinity of 7.120 keV, are still noticeable. Spectral differences of this magnitude are similar in scale to changes induced, for example, by changing the solvent environment of ferrocyanide from water to ethylene glycol[23]. As the sample and solvent in both monochromatic and stochastic measurements are the same, taking such differences at face value and interpreting them as chemical changes in the sample could potentially be misleading. A simple explanation for these differences is that we did not include enough data in the signal recovery algorithm to suitably constrain the solution. Supplementary Figure 1 supports this hypothesis by showing that agreement between monochromatic and stochastic measurements improves in the 7.120 keV region upon the addition of more data. Measurement noise is a major factor in determining how many shots are needed to form an accurate estimate of the true spectrum of the material. One of the advantages of stochastic spectroscopy is that more emission photons are generated per shot. This means that the emission measurement should have a higher signal to noise ratio for the stochastic approach, compared to the monochromatic one. Therefore, it is unlikely that emission measurement noise explains a need for more data in the stochastic measurement. Noise in the SASE diagnostic measurement, i.e., inaccuracy of the reported SASE spectral intensity compared to what the sample experienced, is more likely to be the cause of the problem and may arise due to several noise models. SASE diagnostic noise integrated over the entire SASE bandwidth is added to each shot in stochastic spectroscopy, while only a narrowly filtered portion of it is added to each shot in monochromatic measurements. SASE measurement noise enters both monochromatic and stochastic signal recoveries implicitly when the proposed spectrum is weighed by the SASE spectral diagnostic measurement. Consequently, SASE diagnostic noise is a greater liability to stochastic spectroscopy than monochromatic spectroscopy. We turn to simulations in Fig. 2 to explore the effect of SASE diagnostic noise, as it is not trivial to predict.

Knowing what type of SASE diagnostic noise is impacting stochastic recovery will inform what improvements should be

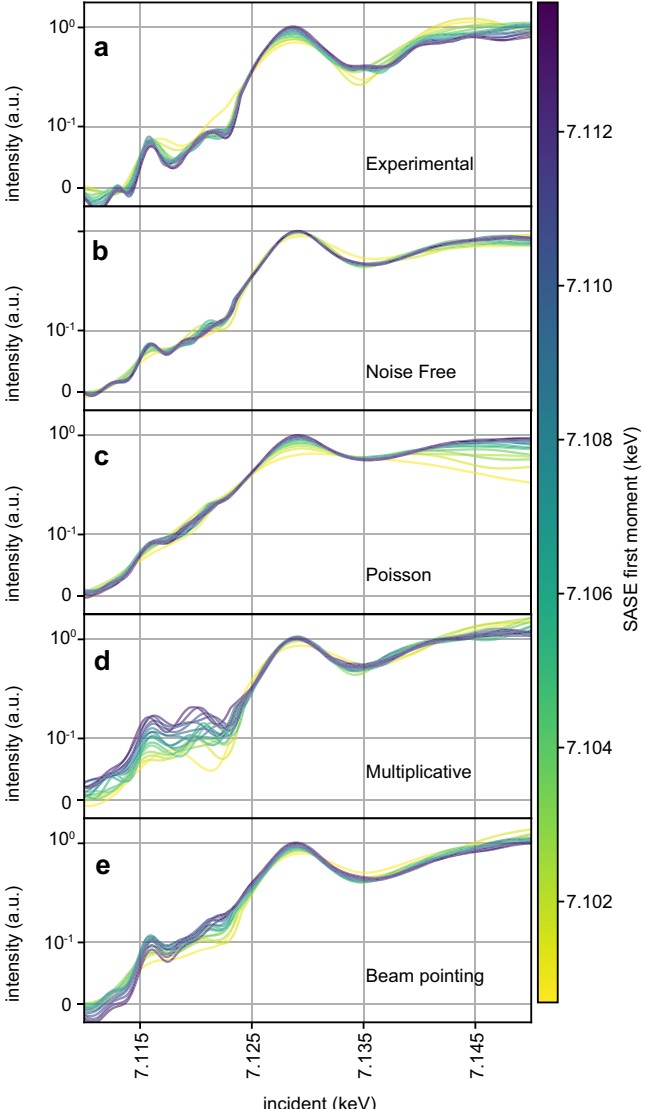

**Fig. 2 Impact of SASE spectral diagnostic inaccuracy.** In each panel, we plot the Fe HERFD absorption spectra fit to 17 independent subsets of the data, each containing 10k shots. The spectra are color coded by the average first moment of the SASE spectral diagnostic used for each independent fit (color bar in units of keV) (see Supplementary Fig. 3a). The spectra are scaled with a symmetric log function to enhance dynamic range of the plot and show variance of the fit in both pre-edge and above edge regions. Panel **a** is fit to experimental data. Panels **b–e** are fit using synthetically generated emission signals created using the same SASE spectral intensity measurements used in panel **a**. In panel **b**, no additional noise is added to the SASE spectral measurements during the reconstruction. In panel **c**, shot noise is added to the SASE spectral diagnostic such that around 330 photons are distributed over 200 pixels in each shot. In the actual experiment, each shot has around 5–10× more photons than this, so the effect shown here is exaggerated to highlight the effect this kind of noise has. In panel **d**, the SASE spectral diagnostic is corrupted by a random, smooth multiplicative filter to simulate spectra distorted by intensity profile fluctuations. In panel **e**, a Gaussian intensity filter of fixed width, but with mean that depends linearly on the SASE first moment is applied to simulate beam pointing that changes slightly with central photon energy.

sought in the method used to measure the SASE spectral intensity as well as in signal recovery algorithms. Two types of SASE diagnostic noise are anticipated: shot noise resulting from low photon collection per pixel and complications arising from spatial

beam profile coupling to spectral amplitude, also called spatial-chirp. While high photon flux is a strength of the SASE spectrometer used, errors in spectral intensity above the edge where spectral intensity drops can have a strong impact on the inferred spectrum. The spectrometer employed in the experiment is known to be sensitive to spatio-spectral variation within the beam profile[7]. Sun et al.[24] showed the XFEL spatial profile for a given X-ray photon energy varies shot to shot. Spatio-spectral variations are thus a known, yet unobservable effect that can potentially alter the correlation between emission and SASE spectral intensity measurements. To probe the effect of these physical noise models, we plot the solution resulting from independent fits to independent subsets of experimental and simulated data in Fig. 2. The data subsets for each fit have been selected so that the average first moment, or central photon energy, of the SASE spectral intensity increases from one subset to the next. Sorting the data subsets this way introduces a systematic change in the underlying SASE intensity diagnostic signal entering each fit, which enhances changes that noise processes on the SASE diagnostic introduce. The line color in Fig. 2 encodes the average SASE diagnostic first moment for the dataset employed in the fit (see color bar). See Supplementary Figure 3a for a comparison of the SASE subsets employed. On the other hand, the ~10 eV range of SASE first moments is small compared to the ~30 eV bandwidth of the SASE beam, so these subsets should all produce very consistent results, except where the SASE spectral intensity tails off. Indeed, one can see in panel b of Fig. 2, which examines synthetic data in which no noise was added to the SASE diagnostic, that there is no clear dependence of the solution on the SASE first moment. In stark contrast to the noiseless SASE diagnostic case, the experimental data in panel a and all simulated noise models, show marked dependence of the solution on the SASE first moment.

The synthetic emission data considered in panels b–e of Fig. 2, are generated using the same subsets of SASE spectra used in panel a in conjunction with a reference spectrum. See Methods for more details on signal simulation. Panels b–e of Fig. 2 all share the same emission responses with identical noise. The distribution of emission noise added in these simulations is depicted in Supplementary Fig. 3b. For Fig. 2c–e, the SASE spectra used during signal reconstruction (but not signal generation) are corrupted by noise. We used shot noise (Poisson) for panel c, multiplicative noise for panel d, and beam pointing noise in panel e. Shot noise produces substantial deviation of the solution as a function of first moment above the edge, but little first moment dependent deviation near the pre-edge. Poisson noise does reduce contrast of weak features, but this lack of weak feature contrast is not seen in our experimental data. Multiplicative noise, which simulates a fluctuating and randomly rugged beam intensity profile, induces strong solution dependence on SASE first moment for all incident energies. Beam pointing noise produces smoother deviations than multiplicative noise, but likewise impacts the entire spectrum. Multiplicative and beam pointing noise are a better match to the behavior of experimental data in panel a. Furthermore, multiplicative noise induces both new peaks and peak position fluctuations in the pre-edge and near-edge regions, which may explain some of the discrepancy between monochromatic and stochastic measurements. Both beam pointing noise and multiplicative noise are a consequence of the spatial-spectral coupling of the SASE spectral diagnostic method employed[7].

So far, we have assumed that the signal generated in the stochastic experiment is linearly related to the spectral intensity of the SASE spectrum. As the SASE pulse is 20–100 more intense than what a monochromator can deliver, non-linear effects may explain the discrepancy between the monochromatic and stochastic measurements. In Kayser and co-worker's[10] stochastic

spectroscopy study of Iron nanoparticles[10] the shape of spectra depended strongly on the X-ray intensity used, which was interpreted to be caused by a variety of non-linear effects. A recent study on X-ray emission of metallic Iron by Alonso-Mori and co-workers[25] also finds large spectral shape changes over a similar range of intensities to that used in[10] as well as in this study. In contrast to spectra of metallic Iron, Alonso-Mori et al.[25] show that spectra of dilute aqueous Iron salt solutions (200 mM), like those used in this study, are much less sensitive to intensity and better preserve valence state information present in Kβ emission spectra. Consequently, we expect non-linear effects to be small in our data, compared to what was seen in locally dense Iron nanoparticles[10].

**Impact of SASE statistics and detection noise**. Supposing the SASE diagnostic can be made as accurate as possible, the distribution of spectral amplitude in a SASE pulse and the noise that affects the emission signal detection are the two main controllable parameters of the stochastic spectroscopy experiment. Emission detection noise depends mainly on sample concentration and the solid angle that the emission spectrometer can collect. Detector technology affects the emission detection noise, but modern X-ray detectors are very close to shot noise limited performance. The distribution of SASE spectral amplitude can be controlled to some degree by various XFEL facility parameters. For example, shorter pulses are comprised of few, broad spikes, while longer pulses are comprised of many well resolved sharp spikes. How many spikes are resolved is also affected by the resolution of the SASE diagnostic spectrometer. As a proxy for many different facility parameters, we characterize a collection of synthetically generated SASE shots by their mutual coherence[26], the maximum absolute value of the cross-correlations between SASE spectra. The more similar the shots are to one another, the higher their mutual coherence. Generally, SASE spectra composed of well separated and sharp spikes produce lower mutual coherence. The impact of mutual coherence and emission noise on the reconstruction accuracy is shown in Fig. 3 through a pure simulation study. The simulation imitates an experimental setup where the SASE beam is split into two equal intensity beams. One beam is monochromatized before striking the sample, while the other is sent directly to a separate identical sample. The incident light spectrum impinging the sample is recorded for both beam paths along with emission signal generated at their respective samples, producing two datasets that are related through their mutual source. We are interested in knowing which arm of the simulated experiment produces a smaller root mean squared error (RMSE) with respect to the ground truth given a fixed measurement time budget.

SASE spectra used in the simulation are synthetically generated. The spectral intensity at each energy is treated as statistically independent of one another and is drawn from a Gamma distribution of unit scale. The shape parameter of the Gamma is adjusted to create datasets with different mutual coherence. Emission signals are produced using the generated SASE spectra as input along with a ground truth spectrum taken from reference data, see Methods for more detail. Emission noise is controlled by setting the number of photons in each beam, so that efficiencies of the mono and collection optics are considered appropriately for each branch of the experiment. Read noise equivalent to 1/5 of a photon in addition to shot noise (Poisson–Gaussian noise) is added to the emission signal used in Fig. 3 panel a, while purely shot noise (Poisson noise) limited measurements are used in panel b. For each condition of SASE mutual coherence and photon budget, the spectrum is reconstructed independently for both monochromatic and full bandwidth experiments. The RMSE of these reconstructions were calculated and we plot in each pixel the ratio of the RMSEs: monochromatic RMSE divided by polychromatic RMSE. When the ratio is greater than one, this indicates that the monochromatic measurement had a larger RMSE (poorer fit) than the polychromatic beam, which is shown as white to red colors in the plot. When the ratio is less than one, the polychromatic beam RMSE is greater and this is depicted with hatched pixels that have white to blue colors.

The vertical black dashed line in Fig. 3 indicates the experimentally measured mutual coherence for the data used in Fig. 1. In compressed sensing[27], which examines imposed projections of a signal, lower mutual coherence between the projections produces better quality results. We see this trend reflected here as superior performance of the stochastic recovery compared to the raster scanned monochromatic experiment. One takeaway of this simulation is that lowering mutual coherence of the SASE spectrum is a way to obtain a measurement time advantage over monochromatic measurements. Conversely, even with ideal SASE diagnostic measurements, at the present SASE mutual coherence, we can expect at best similar measurement times between the two approaches when the per-shot signal to noise is high. Additionally, these simulations show that the stochastic approach is competitive for a wide range of signal to noise values. In fact, both simulations show that the stochastic approach can be advantageous over a broader range of mutual coherence values at lower signal to noise than is the case for higher signal to noise measurements. The presence of a Gaussian (read) noise component in the measurement, as shown in Fig. 3a, improves the relative advantage that stochastic spectroscopy provides over monochromatic spectroscopy, particularly for low concentration samples. Thus, we anticipate stochastic spectroscopy, with an accurate SASE diagnostic, can make challenging time resolved RIXS experiments on dilute systems more feasible.

## Discussion

We have shown experimentally that stochastic spectroscopy can yield similar spectral information content in RXES and HERFD spectroscopies to the more established monochromatic approaches used to collect the same spectra. With the currently available SASE spectral intensity diagnostic, we found that more shots are needed in the stochastic approach to achieve a signal to noise comparable to the monochromatic approach. At the same time, our results suggest that stochastic spectroscopy could be employed in its present state for any sample where monochromatic studies are already feasible and can tolerate small degradation of the signal to noise. Simulations in Fig. 3 show that the relative performance of stochastic spectroscopy is largely unaffected by the signal to noise of each shot, so lower concentration samples will fare similarly to the same sample measured with a monochromatic approach. For very dilute samples, Fig. 3 also indicates that stochastic spectroscopy will offer a measurement advantage once the signal's amplitude is comparable to the read noise of the detector. The relative inefficiency of stochastic spectroscopy that we have observed is unexpected when measurement noise is present only on the emission signal, as we show via simulations in Fig. 3. Some of the issues with how quickly the solution converges to the right answer can be explained by noise on the SASE spectral intensity diagnostic. In analysis presented in Fig. 2, we find that noise models which invoke the spatial-spectral coupling of the SASE spectral diagnostic best reproduce the behavior seen in experimental data, which points to spatial-spectral coupling being a problem that should be addressed in future studies. The strong performance dependence of stochastic spectroscopy on the SASE beam mutual coherence, illustrated in Fig. 3, implies that different XFEL accelerator control parameters

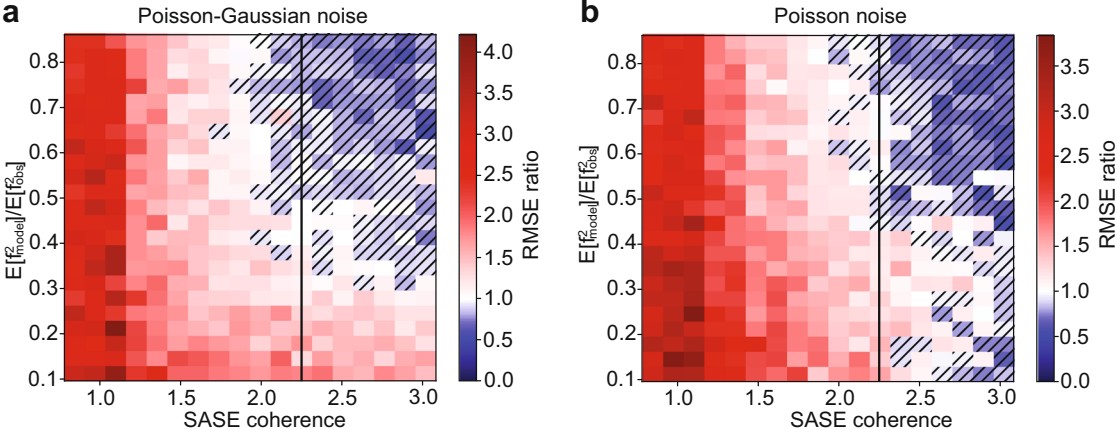

**Fig. 3 Relative performance of stochastic spectroscopy to monochromatic spectroscopy.** We examine the merits of stochastic signal recovery to monochromatic signal recovery as a function of detector noise type (panels), amount of detector noise (vertical axis), and a similarity metric of SASE shots (horizontal axis). Poisson or shot noise is used in panel **b**, while in panel **a**, shot-noise plus Gaussian read noise (Poisson–Gaussian) is used. No noise is added to the SASE spectral diagnostic. The amount of detector noise applied is determined by a ratio of the 2nd moment of the noiseless signal to the noisy signal. A ratio of 1.0 indicates that the detector noise model contributes negligibly to the 2nd moment (a high quality signal), while a ratio of near zero indicates a measurement dominated by detector noise. SASE coherence, defined in the text, is smaller for collections of shots which are less similar (have smaller inner product). Each pixel in panels **a** and **b** depicts a ratio: the RMSE for signal recovered using monochromatized light divided by the RMSE of the signal recovered using the full SASE bandwidth. Each pixel represents the median RMSE ratio of 10 simulated experiments, with 100k shots each. The black line on each image shows the experimentally observed coherence of the SASE light in Fig. 1 as a point of reference.

may be used to improve the performance of stochastic spectroscopy.

Despite some of the challenges exposed in this study, the use of beam diagnostic measurements to characterize a chaotic source in lieu of source control is liberating for XFEL experimental design. The freedom to use the full SASE bandwidth will enable new multi-modal signal collection experiments, like combined scattering and resonant spectroscopy experiments. Additionally, resonant scattering experiments making use of the natural SASE fluctuations in a similar fashion to the experiment discussed here have been recently studied theoretically[28]. A compelling experimental use-case[18], used the natural SASE fluctuations in combination with photo-electron yield to measure X-ray absorption from an attosecond pulse. Here stochastic spectroscopy enables attosecond time-scale optical pump, resonant X-ray probe experiments. The conventional approach to absorption, employing a monochromatized X-ray probe compromises either time resolution or frequency resolution due to the time-bandwidth limit on the X-ray pulse. While similar advantage can be accomplished via a traditional Fourier Transform approach, the natural spectral fluctuations are already present and quick to exploit as little extra instrumentation is needed. The ready availability of natural spectral and temporal fluctuations in the SASE beam have also inspired theoretical studies into more complex multidimensional spectroscopies beyond RXES. Stimulated emission, rather than spontaneous emission, correlated with SASE spectral intensity diagnostics can be used to study a nonlinear form of RXES[29,30]. Combining spectral analysis of diffraction signals and temporal intensity diagnostics have also been explored theoretically[31] to measure time resolved diffraction signals. While this work does not cover these non-linear X-ray scenarios, the use of source diagnostics in correlation with an observable is a common thread. Some of the challenges studied here will be of value in bringing these exciting extensions of XFEL science to mainstream use.

## Methods

**Data collection**. The physical process that produces incident energy-dependent X-ray emission is depicted using an energy level diagram in Fig. 4a. The experimental setup we employed to measure RXES can use either monochromatized or full bandwidth SASE light, so that all other components can remain fixed in either source configuration, allowing us to directly compare the approaches. The setup is summarized in Fig. 4b and was implemented at Beamline 3[32] of SPring-8 Angstrom Compact free-electron Laser (SACLA)[33]. Aside from the offset mirror/monochromator, the setup consists of three main devices: a spectrometer to measure the spectral amplitude of the SASE light, a spectrometer to measure X-ray emission, and a liquid jet to replenish the sample after each XFEL shot. For both spectrometers, the X-ray light was detected with a multiport charge-coupled device[34], a direct-detection pixel array detector capable of single photon registration. The 1s2p emission line from Iron is monitored with an energy dispersive von Hamos spectrometer with ~1 eV resolution and a sampling rate of about 0.6 eV per pixel. The von Hamos spectrometer used a cylindrically bent Ge 440 crystal with 250 mm bending radius. The SASE spectrometer[9] samples the beam after the monochromator/offset mirrors, via a silicon grating, and resolves the spectrum of the light sent to sample with ~0.5 eV resolution and a sampling rate of 0.6 eV per pixel. The SASE spectrometer, also serves as a beam intensity monitor, or $I_0$, for the monochromatic measurement. The crystal analyzer in the SASE spectrometer (a flat Si 220 crystal) trades off spectral resolution in exchange for the ability to resolve the entire SASE spectral bandwidth so that all SASE fluctuations influencing the emission intensity are captured. This type of SASE spectrometer provides a strong signal with little photon noise. The XFEL beam was focused by a Kirkpatrick-Baez mirror system[35] to a ~1 micron spot onto a 200 micron diameter liquid jet. The liquid jet used a nozzle and pump system designed at SACLA[36] and was housed in a Helium chamber also developed there[37]. The liquid jet delivered an aqueous solution of 200 mM Potassium ferrocyanide ($K_4Fe(CN)_6$) via a recirculating reservoir.

**Noise treatment**. Von Hamos emission spectrometers, as well as other hard x-ray spectrometers, capture only a small solid angle of the emission that radiates spherically from the sample. The setup had a collection angle of 67 micro-steradian per eV, resulting in average counting rates of a few photons per pixel (~50 photons per shot in the focused region of the von Hamos at the $K\alpha_1$ peak). Modern X-ray pixel array detectors[38] have very low read noise compared to the signal of a single X-ray photon, less than 1/5 of a photon, so X-ray emission from the sample is dominantly corrupted by Poisson noise. In contrast, the collection solid angle per eV of the SASE spectrometer is more closely matched to the source divergence and directly measures a (~2% at 10 keV) sample of the beam so each shot has hundreds of photons per pixel. Thus, the SASE signal diagnostic should have substantially higher signal to noise, considering only the photon counting statistics. For this reason, we treat the SASE spectral measurement as noise free in the signal recovery algorithm. Statistical inference of the model is also substantially easier when the SASE diagnostic signal is treated as noise-free. However, we saw in the Results that the SASE spectral diagnostic measurement is not actually noise-free and so properly addressing the noise in signal recovery is an avenue for future improvement.

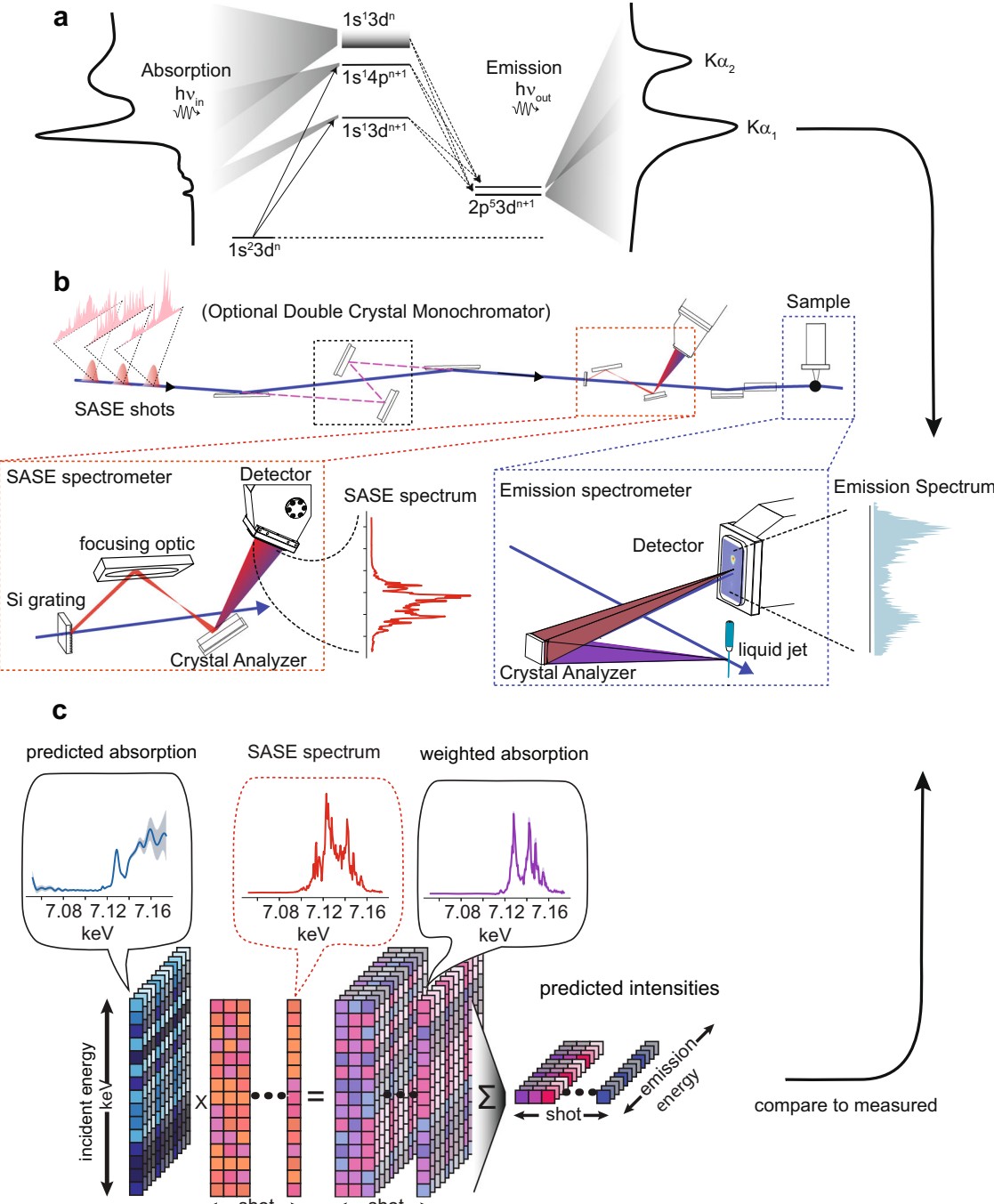

**Fig. 4 Setup and signal generation overview.** A cartoon in panel **a** depicts the atomic transitions involved in K-edge absorption and 1s2p emission of Iron. The beamline configuration used for data collected is shown in panel **b** and shows how the beamline can switch between full bandwidth SASE and monochromatized SASE upstream of both spectrometers. Details of the SASE spectrometer and emission spectrometer are highlighted. The SASE spectrometer works by sampling a collimated beam using a silicon grating and then dispersing the spectral content of the beam onto a pixel array detector via a flat Si 220 analyzer crystal coupled to a reflective focusing optic. A von Hamos configuration is used for emission detection, wherein a single cylindrically bent crystal collects sample emission perpendicular to the beam, re-images, and energy disperses it onto a pixel array detector. Panel **c** depicts the model used for analysis. For each detected emission energy, a proposed absorption spectrum is weighed by the measured SASE spectrum and then sum reduced to form a proposed emission intensity.

**Extraction of RXES spectrum**. Figure 4c depicts how X-ray absorption and emission of a material are related when measured using broadband SASE light. We can summarize Fig. 4c with a matrix equation:

$$\mathbf{Y} = \mathbf{WS} \qquad (1)$$

Here **S** is the RXES spectrum, with absorption axis forming the rows and emission axis forming the columns. **W** are the SASE spectra, with the spectrum of each shot occupying a row of the matrix and with the number of rows being much greater than the number of columns. **Y** is a matrix of X-ray emission intensities, each row consisting of a single-shot Kα emission spectrum. The SASE spectra act as a linear operator that transforms the RXES spectrum to a new space, similar in spirit to the action of the Fourier operator in Fourier spectroscopy. All our simulations of emission $\mathbf{Y}_{sim}$ using SASE light in Figs. 2 and 3 are done using a given spectrum **S** and supplied SASE weighs **W**, according to Eq. (1). The RXES spectrum can be obtained through the approximate inverse transformation: $\hat{\mathbf{S}} = \mathbf{W}^{+}\mathbf{Y}$, where $\mathbf{W}^{+}$ is the Moore-Penrose pseudo-inverse. This solution minimizes the least-squares error

between observations **Y** and the model **WS**. However, measurement noise, arising from low photon counting rates, detector read noise or other effects, cause this solution, which is one of infinitely many in such an over-determined problem, to have high frequency content, to have non-physical negative intensities, and generally to be unsatisfactory.

**Gaussian process regression**. To mitigate the impact of noise and prevent unrealistic solutions, some constraint needs to be imposed on the RXES spectrum being estimated. We may, for example, desire the solution to be smooth and positive. A popular and computationally cheap method to constrain the solution is to penalize the least squares objective by adding a scaled L2 norm, L1 norm, or a more general norm of the solution. The strength of the penalty is a hyper parameter that is determined through grid search[39], Bayesian optimization[40], or other methods[41]. The spectra recovered by Kayser and co-workers[10] used L2 penalization tuned by the L-curve method[41]. We used Gaussian Process Regression (GPR)[42], a Bayesian approach that is like penalized least squares in some regards but allows for the penalty to be optimized through gradient descent on an objective function. GPR also supplies an error estimate of the fit and does this by assuming the RXES spectrum is a functional distribution over many possible spectra, with the distribution on a finite set of energy points taking the form of a multivariate normal distribution. When conditioned on observed data, the conditional distribution becomes tighter, modeling how our certainty in the spectrum improves with added data. We obtain the gradient of the GPR objective function, called the marginal likelihood, through automatic differentiation in Tensorflow[43] and use GPFlow[44] as a basis for our code[45]. Another attractive attribute of GPR is that it fits a function to the data, as opposed to a matrix of coefficients at fixed energy values, which allows one to easily interpolate the response to unmeasured energy points. It is important to note that there are many ways to approach the signal recovery problem. For example, in the work of Driver and co-workers[18], the solution was controlled for smoothness, sparsity in a basis set, and non-negativity through a more general optimization algorithm called Alternating Direction Method of Multipliers[46].

**Signal extraction design space**. A GPR model defines the prior distribution of the RXES spectrum to be $S(x) \sim N(0, k[x, x'])$, where $k[\mathbf{X}, \mathbf{X'}] \stackrel{\text{def}}{=} \mathbf{K}$ is a matrix-valued function called a kernel that maps a collection of energy points $\mathbf{X}$ to a positive definite matrix. The mean solution of GPR is the same as the solution to the penalized least squares objective[47] $\frac{1}{2}\sigma^{-2}(\mathbf{y} - \mathbf{K\alpha})^T(\mathbf{y} - \mathbf{K\alpha}) + \frac{1}{2}\boldsymbol{\alpha}^T\boldsymbol{\alpha}$, where $\boldsymbol{\alpha}$ is the solution. This shows that, in the context of penalized least squares, the kernel defines a generalized norm that we are penalizing. Consequently, the functional form of the kernel plays an important role in the behavior of the optimal solution. In this work, we use only simple stationary kernels that smooth the solution uniformly for all energies, leaving more complex (energy-dependent) options[48] to future studies. GPR assumes the data is corrupted by Gaussian noise, which is very computationally convenient, if not representative of the detector physics. With the Gaussian noise assumption, the solution is analytic given hyper-parameters that are themselves optimized via a quasi-Newton solver from the scipy library[49]. A derivation of the predictive RXES distribution and objective function is supplied in Supplementary Note 1. Because the measurements are repeated at the same energies for each shot, the cost of optimizing the GP is independent of how many shots are measured, but it is cubic in the number of energy points (the cartesian product of incident and emission energies).

## Data availability

The data that support the findings of this study are available at https://doi.org/10.17605/OSF.IO/2CQMZ, hosted by the Center for Open Science.

## Code availability

Code used to replicate the figures can be found at: https://doi.org/10.5281/zenodo.4680002, available for use according to the MIT license, the details of which are included in the repository. Instructions for using the code are supplied. A docker container is included to replicate the software dependencies. The repository contains scripts (python notebook files) which explain and generate figures given pre-processed data that can be separately download (see Data Availability).

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

## Acknowledgements

F.D.F and this work was supported by the Department of Energy, Laboratory Directed Research and Development program at SLAC National Accelerator Laboratory, under contract DE-AC02-76SF00515, in support of the Panofsky Fellowship. J.K., V.K.Y., and J.Y. thank the support from the Director, Office of Science, Office of Basic Energy Sciences (OBES), Division of Chemical Sciences, Geosciences, and Biosciences (CSGB) of the Department of Energy (DOE) under contract DE-AC02-05CH11231. The National Institutes of Health (NIH) provides funding through Grants GM126289 (J.K.), GM110501 (J.Y.), and GM055302 (V.K.Y.). The experiment was performed at SACLA with the approval of Japan Synchrotron Radiation Research Institute (JASRI; Proposal No. 2017A8050). We thank Robert Schoenlein for valuable comments during the review and preparation of this manuscript.

## Author contributions

F.D.F., M.Y., T.K., U.B., J.K., V.K.Y., P.W., and J.Y. planned and designed the experiment; F.D.F, T.T., D.Y., Y.L., T.F., K.U., M.Y., T.K., T.C.W., R.A.M., J.K., V.K.Y., P.W., J.Y., executed the experiment, F.D.F., A.L., T.T., D.Y., Y.L., and T.F. analyzed the data. F.D.F., U.B., J.Y. and P.W. wrote the manuscript with input from all authors.

## Competing interests

The authors declare no competing interests.
