## [Peer Review File · Communications Chemistry]

Reviewers' comments:

Reviewer #1 (Remarks to the Author):

The manuscript entitled "Multidimensional spectroscopy from broadband stochastic pulses at an X-ray free electron laser" is a nice work about performing Resonant inelastic Raman scattering (RIXS) with self-amplified spontaneous-emission (SASE) pulses in a dispersive approach without using a monochromator but rather by using the measured shot-to-shot SASE spectrum.

The paper is well written and presents interesting results towards the development of "stochastic RIXS spectroscopy". However, the manuscript focuses on the technique rather than in a particular interesting problem in Chemistry. Also, the presented results do not demonstrate the potential of stochastic spectroscopy beyond current spectroscopy techniques. Although the work is very interesting, it falls outside the scope of Communication chemistry research and is instead recommended for a much more specialized technical journal instead.

Some comments for the authors:

i) I would recommend to change the title, the multidimensional word is confusing.

ii) I would expect that if one excites a resonance with two different modes (photon energies), one may produce some interferences at the first-order perturbation theory that would depend on the detuning and the stochastic phase of the modes. Perhaps the authors could add a paragraph discussing this point.

iii) Other recent works take advantage of dispersive XAS approaches with SASE pulses, as Ref. 10 and 18, as mentioned in the manuscript. It would be worthwhile to add a paragraph to mention similarities or differences between the used methods to correlate the data.

iv) The experiments have been carried out using a very high concentration of a commercially purchased compound, ferrocyanide. It would be interesting to remark how the approaches discussed in the paper would benefit more realistic designer systems typically measured in much lower concentrations and employed in catalysis or spin crossover applications.

Reviewer #2 (Remarks to the Author):

In their paper "Multidimensional spectroscopy from broadband stochastic pulses at an X-ray free electron laser," Fuller et al. provide two-dimensional spectra of the Fe K pre-edge region of aqueous ferrocyanide (Fe) with stochastic x-ray free-electron laser (FEL). The spectra are inferred from the fluorescence signal emitted by the sample and the stochastic X-ray spectrum, and highlight the correlation between incident and scattered frequencies. The technique was previously applied by Kayser et al. in ref. [10] to obtain the Fe K-edge x-ray absorption spectrum of Fe₂O₃. In contrast to that work, the technique is here applied to access the K pre-edge of aqueous ferrocyanide. The Fe K

pre-edge region is chemically more relevant than the K-edge, since it is sensitive to the properties of the ligand to which Fe is bound.

The paper is well written and scientifically sound. It represents a significant step towards the application of multidimensional nonlinear x-ray spectroscopy with FEL pulses. Most FEL sources are based on the self-amplified spontaneous emission (SASE) principle, which produces stochastic pulses with a fluctuating temporal and spectral intensity. While the coherence properties of X-ray SASE FEL pulses can be improved, e.g., by monochromators, this is accompanied by a loss of intensity. This can compromise the applicability of nonlinear spectroscopy techniques, for which a higher intensity is advantageous. Stochastic SASE FEL pulses thus provide the necessary pulse intensity needed for such applications, but the relevant spectroscopy information has to be extracted from the chaotic signals thereby generated. The authors demonstrate that suitable algorithms can provide two-dimensional spectra comparable with those obtained with monochromatized pulses, as confirmed experimentally by repeating the measurement with a SASE beam and a monochromatized beam.

The authors reconstruct the two-dimensional correlation of incident and scattered frequencies in fluorescence. This fluorescence signal is sequential, in contrast to RIXS, which is a Raman non-sequential signal. I therefore recommend the authors to refrain from referring to their signal as RIXS, as also done in the previous ref. [10]. This does not diminish the relevance and impact of this work, which is an impressive demonstration of multidimensional nonlinear x-ray spectroscopy with stochastic FEL pulses.

The data shown in Figure 2 confirm that the technique applied with stochastic FEL pulses is in general in agreement with measurements based on monochromatic pulses, even though with small differences in the position of the peaks and in the contrast. I particularly appreciated the analysis in Figure 3, where the impact of noise due to the SASE spectral diagnostic is investigated. In such case, the authors used synthetic data, where different realistic types of detector noise were added artificially, and studied the effect on the reconstruction method. Detector noise can obstacle the application of correlation-based approaches with stochastic x-ray FEL pulses. It is thus important to properly quantify this effect, so that it can be suitably counteracted.

I found the manuscript very thorough and clear, and I recommend it for publications in Communications Chemistry. In addition to not referring to their spectra as to the RIXS spectra, the authors should better explain how the synthetically generated SASE spectra used in Fig. 4 were simulated. The spectra are characterized by their coherence properties, and I realize that the details of the pulse

simulation algorithm may be minor. Yet, I think it would be helpful if this was better clarified.

Reviewers' comments:

Received commentary from reviewers are shown in *red italics*. Author's response to reviewer's comments are shown in normal black font.

Reviewer #1 (Remarks to the Author):

The manuscript entitled “Multidimensional spectroscopy from broadband stochastic pulses at an X-ray free electron laser” is a nice work about performing Resonant inelastic Raman scattering (RIXS) with self-amplified spontaneous-emission (SASE) pulses in a dispersive approach without using a monochromator but rather by using the measured shot-to-shot SASE spectrum.

The paper is well written and presents interesting results towards the development of “stochastic RIXS spectroscopy”. However, the manuscript focuses on the technique rather than in a particular interesting problem in Chemistry. Also, the presented results do not demonstrate the potential of stochastic spectroscopy beyond current spectroscopy techniques. Although the work is very interesting, it falls outside the scope of Communication chemistry research and is instead recommended for a much more specialized technical journal instead.

We thank the reviewer for their valuable comments. Regarding the suitability of our work to be published in Communication Chemistry, we acknowledge that the manuscript is focused on the new technique more so than a particular problem in the field of chemistry. However, our results and analysis are significant because they show that RIXS/pre-edge signals can be collected at X-ray free electron lasers using stochastic spectroscopy, which was previously not well demonstrated or explored. RIXS and pre-edge signals are of interest to the broad chemistry community and have found established usage within the field of inorganic and bioinorganic chemistry. Our report is of foundational interest to scientists in the field of chemistry who desire to use RIXS/pre-edge signals, laying out what one can expect from stochastic spectroscopy in terms of signal to noise, measurement time, and potential sources of experimental error. Given this groundwork, it will be easier for chemists to propose and explore some of the new directions that stochastic spectroscopy can bring, which are highlighted in the introduction as well as in the discussion section.

Some comments for the authors:

i) I would recommend to change the title, the multidimensional word is confusing.

We agree that multidimensional is perhaps overbroad, and will change the title to “Resonant X-ray Emission Spectroscopy from broadband stochastic pulses at an X-ray free electron laser”

ii) I would expect that if one excites a resonance with two different modes (photon energies), one may produce some interferences at the first-order perturbation theory that would depend on the detuning and the stochastic phase of the modes. Perhaps the authors could add a paragraph discussing this point.

We agree with the reviewer that an interference between two electronic states may be created by two different photon energies in a single broadband pulse. As the reviewer rightly points out, the resulting excitation will depend on the relative detuning and phase of these frequency components. Signals of this kind are an active area of investigation, for example to create coherent electronic wave-packets and related effects exploiting the element specificity of X-rays. For example, see: [10.1103/PhysRevLett.125.073203](https://doi.org/10.1103/PhysRevLett.125.073203).

With regards to our measurements, we expect the contribution from interferences to be small. The SASE pulses from an XFEL are partially coherent because multiple radiation modes are being amplified, as indicated in ref [5] of the revised manuscript. Because of this, we refer the reviewer to [10.1063/1.460467](https://doi.org/10.1063/1.460467), which shows that for signals scaling linearly with the intensity of a partially coherent source, the various modes of the radiation cause the oscillatory interference terms to average out within a single pulse. If we consider a non-linear measurement, we will have an ensemble average over various non-linear signals per pulse in which it may be possible for the effect of interferences to survive. However, this means of interferences affecting the overall signal will produce a small effect compared to those generated by a single photon interaction.

As we think the impact of interferences created by the broadband pulse will manifest mainly for non-linear signals, we address the issue broadly as a non-linear effect in the systematic noise section of the revised manuscript. We added a paragraph there which generally discusses why we think non-linear effects are bound to be small in our measurement. Our response to Reviewer #2 offers some more detail on the matter.

iii) Other recent works take advantage of dispersive XAS approaches with SASE pulses, as Ref. 10 and 18, as mentioned in the manuscript. It would be worthwhile to add a paragraph to mention similarities or differences between the used methods to correlate the data.

This is a great point and we thank the reviewer for raising it. We have added two sentences in the Gaussian Process Regression section which make the comparison of our approach more explicit in the context of these two references. In short summary, our method amounts to a different approach to regularization and we view it to be no more or less correct than other approaches in principle. There may be practical advantages to perform the regression one way or another, but in our own study of various approaches, we did not find the computational method of extraction to be as significant as the more experimentally focused aspects that we raise in the manuscript.

iv) The experiments have been carried out using a very high concentration of a commercially purchased compound, ferrocyanide. It would be interesting to remark how the approaches discussed in the paper would benefit more realistic designer systems typically measured in much lower concentrations and employed in catalysis or spin crossover applications.

We thank the reviewer for highlighting this, and we added a comment about this in the first paragraph of the discussion section of the revised manuscript. The fact that we demonstrate stochastic spectroscopy in direct comparison to the monochromatic method on the same sample and with the same analyzer setup and same beamline should answer any question about how stochastic spectroscopy will fare in comparison to the established approach for more interesting low concentration designer systems. If XFEL absorption experiments were feasible at a particular XFEL endstation with the concentrations of interest using a monochromator, then

using a stochastic spectroscopy approach will not substantially degrade that feasibility, as the measurement times were shown to be comparable. Our simulation work in Figure 4 also indicates that reducing the concentration of the sample, and commensurately increasing the per shot measurement noise, does not substantially change measurement time relative to the monochromatic experiment (this would be a vertical slice of the two figures, where obtaining a constant RMSE ratio indicates similar performance across per-shot measurement noise). The main product that stochastic spectroscopy offers chemists is the freedom to explore resonant signals without requiring monochromatic light. We highlight in the introduction and discussion when this experimental freedom can be advantageous. For example, simultaneous absorption and non-resonant scattering and attosecond absorption.

Reviewer #2 (Remarks to the Author):

In their paper “Multidimensional spectroscopy from broadband stochastic pulses at an X-ray free electron laser,” Fuller et al. provide two-dimensional spectra of the Fe K pre-edge region of aqueous ferrocyanide (Fe) with stochastic x-ray free-electron laser (FEL). The spectra are inferred from the fluorescence signal emitted by the sample and the stochastic X-ray spectrum, and highlight the correlation between incident and scattered frequencies. The technique was previously applied by Kayser et al. in ref. [10] to obtain the Fe K-edge x-ray absorption spectrum of Fe₂O₃. In contrast to that work, the technique is here applied to access the K pre-edge of aqueous ferrocyanide. The Fe K pre-edge region is chemically more relevant than the K-edge, since it is sensitive to the properties of the ligand to which Fe is bound.

The paper is well written and scientifically sound. It represents a significant step towards the application of multidimensional nonlinear x-ray spectroscopy with FEL pulses. Most FEL sources are based on the self-amplified spontaneous emission (SASE) principle, which produces stochastic pulses with a fluctuating temporal and spectral intensity. While the coherence properties of X-ray SASE FEL pulses can be improved, e.g., by monochromators, this is accompanied by a loss of intensity. This can compromise the applicability of nonlinear spectroscopy techniques, for which a higher intensity is advantageous. Stochastic SASE FEL pulses thus provide the necessary pulse intensity needed for such applications, but the relevant spectroscopy information has to be extracted from the chaotic signals thereby generated. The authors demonstrate that suitable algorithms can provide two-dimensional spectra comparable with those obtained with monochromatized pulses, as confirmed experimentally by repeating the measurement with a SASE beam and a monochromatized beam.

The authors reconstruct the two-dimensional correlation of incident and scattered frequencies in fluorescence. This fluorescence signal is sequential, in contrast to RIXS, which is a Raman non-sequential signal. I therefore recommend the authors to refrain from referring to their signal as RIXS, as also done in the previous ref. [10]. This does not diminish the relevance and impact of this work, which is an impressive demonstration of multidimensional nonlinear x-ray spectroscopy with stochastic FEL pulses.

We thank the reviewer for their kind words and excellent summary of our work. We respectfully disagree that we should refrain from calling the signal we have obtained “RIXS”, on the basis

that we are presenting data that is dominantly linear (non-sequential), rather than non-linear, and reproduce pre-edge signals. We admit RIXS is more limited in application (referring to the pre-edge region only in many other works) and that it is more appropriate to refer to our entire measurement as RXES “Resonant X-ray Emission Spectroscopy”. Ref [10] likewise refers to their measurement as RXES. For discussion of the pre-edge signals, RIXS is still appropriate. In the revised manuscript we have changed the name to RXES unless specifically referring to the pre-edge region.

The reviewer mentions that RIXS is non-sequential, which is true; RIXS arises from single core-hole fluorescence, as oppose to a sequential ionization process like double core hole fluorescence or from an atom affected by collisional ionization from neighboring atoms. The fact that we used 10 femtosecond pulses with a peak power of $\sim 1\text{E}18$ Watts cm^{-2} does mean that some double core-hole fluorescence signal and other non-linear effects can be present in the data. How much will be present? We refer the reviewer to a recently published study here: [10.1038/s41598-020-74003-1](https://doi.org/10.1038/s41598-020-74003-1) (ref 42 in the revised manuscript). In this study, they examined how various X-ray emission signals are affected by intensity and importantly sample concentration. When studying pure metal samples at $9\text{E}17$ Watts cm^{-2} , very close to our excitation conditions, substantial effect on the emission spectra is observed for 10 femtosecond pulses, causing significant degradation to spectral features associated with the Iron’s valence state, as is best seen in the $\text{K}\beta$ spectra they present. However, when they examined 200 mM sample in water – exactly matching the sample condition we report – and with similar excitation condition to ours, they see dramatically reduced effect as a function of intensity. The authors there explain that this is because the dominant non-linearity seen in metallic iron is collisional ionization from neighboring atoms. Reducing the concentration of iron atoms to 200 mM nearly eliminates effects from neighbors on the femtosecond timescale. In the author’s experience, we can only produce an appreciable amount of sequential ionization on a single atom (10% of the single core-hole fluorescence) with peak powers that are 100x higher than what we used in this study ($1\text{E}20$ Watts cm^{-2}).

In the study by Kayser, ref [10], we want to point out that their sample was nanoparticles of Iron Oxide. So, while the overall concentration of their sample may only be 250 mM ($\text{Fe}_2(\text{III})\text{O}_3$, 159 g/mol, 4% solution by weight), the size of the nanoparticles reported was 30 nm, which with a density per nanoparticle of around 4.8 g/mL would give around $5\text{E}5$ iron atoms packed nearly as densely as metallic iron (which has density ~ 7 g/mL). It may be that some of the non-linear effects they report are related to the density of the sample. Ref [10]’s reported $\text{K}\beta$ emission spectra (Figure 4a) as a function of intensity excited above the edge does closely match the more recently published study (Figure 1b) for metallic iron in trend. We think this difference of sample density could be one major reason why we see the pre-edge more well resolved than in ref 10.

Second, the double core-hole fluorescence signals present in the data should scale quadratically with the incident light intensity. Our model forces the reported response to scale linearly with the incident light intensity and would push quadratic signals into the noise component of the fit. One might argue that non-linearities are then a source of systematic noise, and this is true, but as argued above we believe the effect to be small.

In the revised manuscript, we have added a paragraph to the section on systematic effects which summarizes the discussion above (in fewer words).

The data shown in Figure 2 confirm that the technique applied with

stochastic FEL pulses is in general in agreement with measurements based on monochromatic pulses, even though with small differences in the position of the peaks and in the contrast. I particularly appreciated the analysis in Figure 3, where the impact of noise due to the SASE spectral diagnostic is investigated. In such case, the authors used synthetic data, where different realistic types of detector noise were added artificially, and studied the effect on the reconstruction method. Detector noise can obstacle the application of correlation-based approaches with stochastic x-ray FEL pulses. It is thus important to properly quantify this effect, so that it can be suitably counteracted.

We are very glad that the reviewer appreciates this portion of the manuscript.

I found the manuscript very thorough and clear, and I recommend it for publications in Communications Chemistry. In addition to not referring to their spectra as to the RIXS spectra, the authors should better explain how the synthetically generated SASE spectra used in Fig. 4 were simulated. The spectra are characterized by their coherence properties, and I realize that the details of the pulse simulation algorithm may be minor. Yet, I think it would be helpful if this was better clarified.

We have added two sentences to the revised manuscript describing how this was done.

REVIEWERS' COMMENTS:

Reviewer #1 (Remarks to the Author):

I thank the authors for their response. After carefully reading their answers, I believe my main criticism still stands; the manuscript is still too technical and not clear how this technique will enhance current capabilities at XFELs. This of course does not diminish the quality of the presented work, which I think it is a very interesting work.

Reviewer #2 (Remarks to the Author):

The authors have adequately addressed my comments and remarks, and I recommend publication in Communications Chemistry.